# Interlayer quantum transport in Dirac semimetal BaGa$_2$

Sheng Xu[1,4], Changhua Bao[2,4], Peng-Jie Guo[1,4], Yi-Yan Wang [1,4], Qiao-He Yu[1], Lin-Lin Sun[1], Yuan Su[1], Kai Liu[1], Zhong-Yi Lu[1], Shuyun Zhou [2,3] & Tian-Long Xia [1✉]

The quantum limit is quite easy to achieve once the band crossing exists exactly at the Fermi level ($E_F$) in topological semimetals. In multilayered Dirac fermion systems, the density of Dirac fermions on the zeroth Landau levels (LLs) increases in proportion to the magnetic field, resulting in intriguing angle- and field-dependent interlayer tunneling conductivity near the quantum limit. BaGa$_2$ is an example of a multilayered Dirac semimetal with its quasi-2D Dirac cone located at $E_F$, providing a good platform to study its interlayer transport properties. In this paper, we report the negative interlayer magnetoresistance induced by the tunneling of Dirac fermions between the zeroth LLs of neighboring Ga layers in BaGa$_2$. When the field deviates from the c-axis, the interlayer resistivity $\rho_{zz}(\theta)$ increases and finally results in a peak with the applied field perpendicular to the c-axis. These unusual interlayer transport properties are observed together in the Dirac semimetal under ambient pressure and are well explained by the model of tunneling between Dirac fermions in the quantum limit.

[1] Department of Physics and Beijing Key Laboratory of Opto-electronic Functional Materials & Micro-nano Devices, Renmin University of China, Beijing 100872, P. R. China. [2] State Key Laboratory of Low Dimensional Quantum Physics and Department of Physics, Tsinghua University, Beijing 100084, P. R. China. [3] Collaborative Innovation Center of Quantum Matter, Beijing, P. R. China. [4] These authors contributed equally: Sheng Xu, Changhua Bao, Peng-Jie Guo, Yi-Yan Wang. ✉email: tlxia@ruc.edu.cn

Topological semimetals have attracted tremendous attention due to their novel properties, such as high carrier mobility, large transverse magnetoresistance (MR), non-trivial Berry phase, and chiral anomaly[1–32]. Quantum oscillations, from which the Berry phase, quantum mobility, and effective mass can be extracted, is a routine method in the transport measurement to study topological materials. Obvious Shubnikov–de Haas (SdH) oscillation has been observed in graphene and 3D topological semimetals, e.g. $Cd_3As_2$[7–12], $Na_3Bi$[3–6], TaAs family[13–23,30–32], and $MoTe_2$[24–29]. A particular interesting case is that only the zeroth LL is occupied and no more quantum oscillations should be observed once the quantum limit is achieved. In 2D Dirac/Weyl system, the LL energy of the 2D Dirac/Weyl band can be described as $\varepsilon = \pm v_F\sqrt{2e\hbar B|n|}$ $(n = 0, \pm1, \pm2...)$[33]. Once the Dirac/Weyl point locates at $E_F$, the quantum limit is easily reached when a small field is applied and the zeroth LL is always locked at $E_F$. Thus, the zeroth LL's degeneracy increases dramatically with the field and the Dirac/Weyl fermions on the zeroth LLs participate in the interlayer magnetotransport directly through tunneling mechanism[34], inducing the exotic phenomena of negative interlayer magnetoresistance (NIMR) and interlayer resistivity peak as shown in $\alpha$-(BEDT-TTF)$_2$I$_3$ and $YbMnBi_2$[34–36]. However, the 2D Dirac cone in $\alpha$-(BEDT-TTF)$_2$I$_3$ only exists under pressure and the NIMR is absent in $YbMnBi_2$[34–36].

BaGa$_2$ is recently predicted to be a candidate of Dirac semimetals. The crystal structure of BaGa$_2$ consists of Ga honeycomb-net layers and Ba layers. Its quasi-2D Dirac cone originates from Ga $p_z$ orbitals with the Dirac point located at $E_F$[37], while Ba layer between two Ga honeycomb-net layers act as the interlayer transport barrier. Thus, BaGa$_2$ is believed as an ideal candidate of quasi-2D Dirac material and it provides a good platform to investigate the unusual interlayer transport properties induced by the tunneling of Dirac fermions on the zeroth LLs. Figure 1a presents the schematic illustration of the Dirac fermion interlayer tunneling on the zeroth LLs in BaGa$_2$.

In this work, the quasi-2D Dirac cone at K with the Dirac point located at $E_F$ is confirmed by the first-principles calculations, angle-dependent dHvA oscillations and angle-resolved photo-emission spectroscopy (ARPES) measurements. The above-mentioned NIMR and peak in angle-dependent interlayer resistivity are both observed clearly in BaGa$_2$ for the first time and well explained by the model of Dirac fermion tunneling between the zeroth LLs[34].

## Results

**Materials characteristic and in-plane transport properties.** BaGa$_2$ single crystals are grown by self-flux method (see the "Methods" section). The crystal structure consists of Ba planes and the Ga honeycomb-net planes which shows 2D characteristic. The (00l) lattice plane and lattice parameters are determined by the X-ray diffraction (XRD) measurements on BaGa$_2$ single crystal and powder as shown in Fig. 1b and c. Both the in-plane resistivity ($\rho_{xx}$) and the interlayer resistivity ($\rho_{zz}$) show metallic behavior (Fig. 1d). The ratio of $\rho_{zz}/\rho_{xx}$ is about 26 at 2.5 K, indicating a moderate electronic structure anisotropy. The measurements of Hall resistivity $\rho_{xy}$ reveal that the in-plane dominant carrier is hole-like with the carrier concentration and mobility estimated to be $n_h = 4.28 \times 10^{21}$ cm$^{-3}$ and $\mu_h = 3277$ cm$^2$ V$^{-1}$ s$^{-1}$ at 2 K. The MR of BaGa$_2$ reaches 400% and shows no sign of SdH oscillation (Fig. 1e).

**dHvA quantum oscillations.** dHvA oscillations at various temperatures with the magnetic field along [00l] direction is shown in Fig. 2a. Four fundamental frequencies $F_\alpha = 36.9$ T, $F_\beta = 56.6$ T, $F_\gamma = 356.0$ T, $F_\eta = 1862.0$ T are confirmed to exist after fast Fourier transform (FFT) analysis of the oscillatory component $\Delta M$ as shown in Fig. 2b and c.

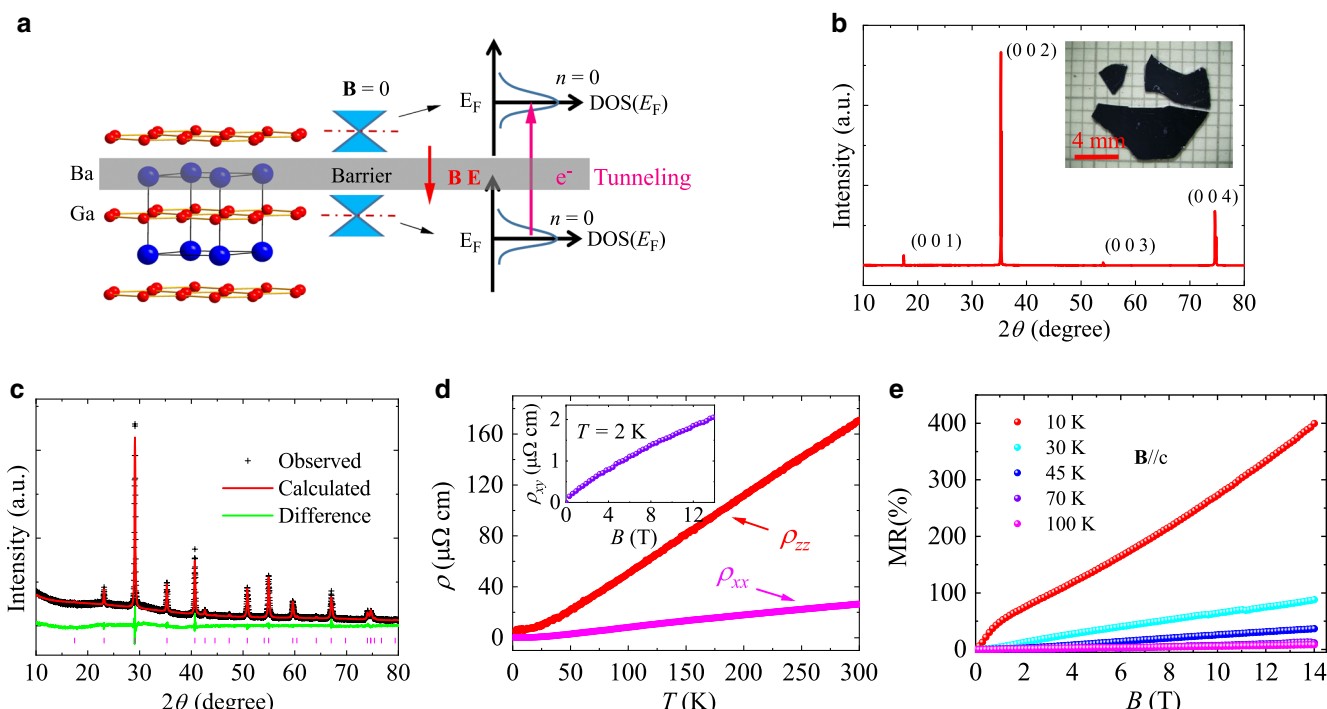

**Fig. 1 Crystal structure and in-plane transport properties of BaGa$_2$. a** Crystal structure and schematic of the interlayer tunneling of zeroth LLs's Dirac fermions of BaGa$_2$. The red and blue balls represent Ga and Ba atoms, respectively. **b** Single crystal X-ray diffraction pattern. Inset: the picture of BaGa$_2$ crystal. **c** Powder X-ray diffraction pattern with refined lattice parameters $a = 4.43$ Å and $c = 5.08$ Å (SG: P6/mmm). **d** Temperature dependence of in-plane $\rho_{xx}$(T) and interlayer $\rho_{zz}$(T). Inset: field dependence of Hall resistivity $\rho_{xy}$ at 2 K. **e** In-plane MR at different temperatures.

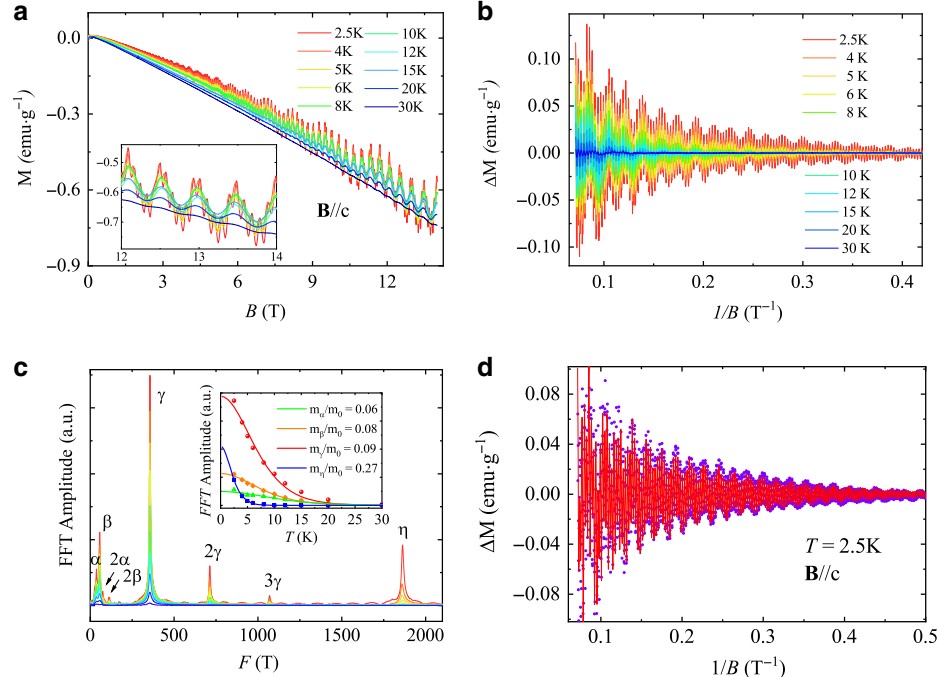

**Fig. 2 dHvA oscillations of BaGa₂. a** The dHvA oscillation with **B**//c. **b** The oscillatory component $\Delta M$ as a function of $1/B$ after subtracting a smoothing background. **c** FFT spectra of $\Delta M$. Inset: The effective masses fitted by the thermal factor of LK formula. **d** The LK fitting (red line) of $\Delta M$ at 2.5 K.

**Table 1 Parameters extracted from dHvA oscillations.**

| Peak | $F$ (T) | $T_D$ (K) | $m^*/m_0$ | $\tau_q$(ps) | $\mu_q$ (cm² V⁻¹ s⁻¹) | $\phi_B$ | $\upsilon_F$ (10⁶ ms⁻¹) | $k_F$ (Å⁻¹) | $E_F$(eV) |
|---|---|---|---|---|---|---|---|---|---|
| $\alpha$ | 36.9 | 27.3 | 0.06 | 0.045 | 1334 | $1.39\pi$ | 0.66 | 0.033 | 0.144 |
| $\beta$ | 56.6 | 8.0 | 0.08 | 0.152 | 3424 | $1.73\pi$ | 0.62 | 0.041 | 0.168 |
| $\gamma$ | 356.0 | 5.9 | 0.09 | 0.210 | 4107 | $0.75\pi$ | 1.37 | 0.104 | 0.937 |
| $\eta$ | 1862.0 | 7.2 | 0.27 | 0.168 | 1179 | $1.37\pi$ | 1.02 | 0.238 | 1.597 |

$F$ is frequency of the dHvA oscillations; $T_D$ is Dingle temperature; $m^*/m_0$ is the ratio of effective mass to free electron mass; $\tau_q$ is quantum relaxation time; $\mu_q$ is quantum mobility; $\phi_B$ represents Berry phase; $\upsilon_F$, $k_F$, and $E_F$ represent Fermi velocity, Fermi vector, and Fermi energy, respectively.

The dHvA oscillations can be described by the Lifshitz–Kosevich (LK) formula[38]

$$\Delta M \propto -B^\gamma R_T R_D \sin\left[2\pi \times \left(\frac{F}{B} - \frac{1}{2} + \beta + \delta\right)\right] \quad (1)$$

where $R_T = (\lambda T\mu/B)/\sinh(\lambda T\mu/B)$, $R_D = \exp(-\lambda T_D\mu/B)$ and $\lambda = (2\pi^2 k_B m_0)/(\hbar e)$. $\mu$ is the ratio of effective mass $m^*$ to free electron mass $m_0$, and $T_D$ is the Dingle temperature. $\gamma = 0$, $\delta = 0$ for a 2D system, and $\gamma = 1/2$, $\delta = \pm 1/8$ for a 3D system. $\beta = \phi_B/2\pi$ and $\phi_B$ is the Berry phase. The inset of Fig. 2c displays the temperature-dependent FFT amplitudes and fittings using the thermal factor $R_T$ in LK formula. The obtained effective cyclotron masses are quite small, comparable with those in topological semimetals, due to the almost linear dispersion though they all originate from trivial bands lately determined by the magnetotransport analysis and first-principles calculations. In order to extract the Berry phase conveniently, we applied four-band LK formula fitting directly. The fitting result is displayed in Fig. 2d with the violet dots being experimental results and the red line being the fitting curve. All of the Berry phase of these four pockets greatly deviate from the non-trivial value $\pi$ implying that these bands may be trivial. Detailed data is exhibited in Table 1. The angle-dependent dHvA oscillation measurements are applied to further reveal the Fermi surface of BaGa₂ as exhibited in Fig. 3a. Figure 3b and c

shows the FFT spectra of the corresponding dHvA oscillations with the **B** rotating from **B**//c to **B**//ab. The fundamental frequencies increase with the angle $\theta$ and vanish at $\theta = 90°$, indicating the moderate electronic structure anisotropy of these trivial bands. If these trivial bands are highly 2D-like, these frequencies should vanish once the magnetic field rotates by a small angle, which is inconsistent with the experimental measurements that $F_\alpha$, $F_\beta$, and $F_\gamma$ are still observable at $\theta = 56°$. Besides, the twofold symmetry of polar plot MR (Fig. 3d) with **B** always vertical to **I** at 2.5 K and the ratio of $\rho_{xx}(14\,T, 0°)/\rho_{xx}(14\,T, 90°) \approx 5$ also indicate a moderate electronic structure anisotropy in BaGa₂.

**First-principles calculations.** The band structure and Fermi surfaces of BaGa₂ calculated with spin–orbit coupling (SOC) effect included are shown in Fig. 4. As exhibited in Fig. 4a, the Dirac cone locates exactly at K point ($E_F$) in the Brillouin zone (BZ), which is consistent with previous work[37]. In addition, there exist three trivial bands crossing the Fermi level with the corresponding FSs demonstrated in Fig. 4b–d. The hole-type FSs (Fig. 4b and c) are open-orbit FSs along the $k_z$ direction exhibiting quasi-2D characteristic. Notably, different from the ideal cylindrical Fermi surface the lotus-root shape hole-type FS in Fig. 4b has two evident cross sections indicating the existence

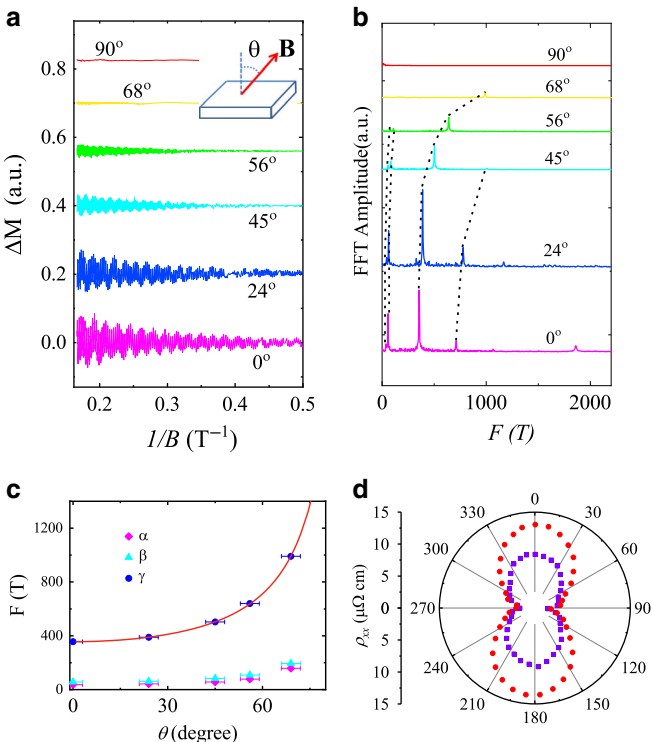

**Fig. 3 Angle-dependent dHvA oscillation and in-plane MR of BaGa$_2$.**
**a** Angle-dependent dHvA oscillations at 1.8 K. **b** Corresponding FFT spectra with $\theta$ varying from 0° to 90°. **c** FFT frequencies as a function of angle. Solid red line is the fitting curve with equation $F = F_0/\cos\theta$. Error bars are estimated from sample holders and defined as s.d. **d** Polar plot of MR at 9 T (violet dots) and 14 T (red dots) with **B** always vertical to **I** at 2.5 K.

of $k_z$ dispersion, which is consistent with the angle-dependent dHvA oscillations (Fig. 3). According to the Onsager relation $F = (\phi_0/2\pi^2) = (\hbar/2\pi e)A$, the frequency $F$ is proportional to the extreme cross section $A$ of FS normal to the magnetic field. The calculated dHvA frequencies from the hole-type Fermi surface Fig. 4b are 61.4 and 430.1 T corresponding to the observed frequencies $F_\beta$ and $F_\gamma$. The calculated frequency from the other hole-type Fermi surface Fig. 4c is 2273.4 T corresponding to the observed frequency $F_\eta$. The calculated largest frequency from the electron-type Fermi surface (Fig. 4d) is 4608.3 T which is too high to be observed at 2.5 K and 14 T. The observed frequency $F_\alpha$ may originate from the electron-type Fermi surface (Fig. 4d) as there is a branch along K–H according to the calculations. Thus, the trivial Berry phases obtained from the fitting are acceptable since they are all from the trivial bands. As for the Dirac band, no frequencies corresponding to the oscillation should be observed since it is located exactly at $E_F$. In addition to the Dirac cone, our first-principles calculations reveal that there is an electron-type pocket along L–H and two hole-type pockets at Γ point in the BZ, which is slightly different from previous results in which there seems to be an extra analogous hole-type pocket at H point and only one hole-type pocket at Γ point[37]. The difference is acceptable considering that different calculation methods are employed. Besides, in real materials the Fermi surface is affected by many factors, such as temperature, magnetic field, defects, etc. According to the calculations, the Dirac point locates at the high-symmetry point K and $E_F$. Although the Dirac point is not protected by the crystal symmetry and opens a small gap of ~10 meV by considering the SOC effect, the low-energy excitation can still be described by the Dirac equation and possesses the characteristic of Dirac fermions.

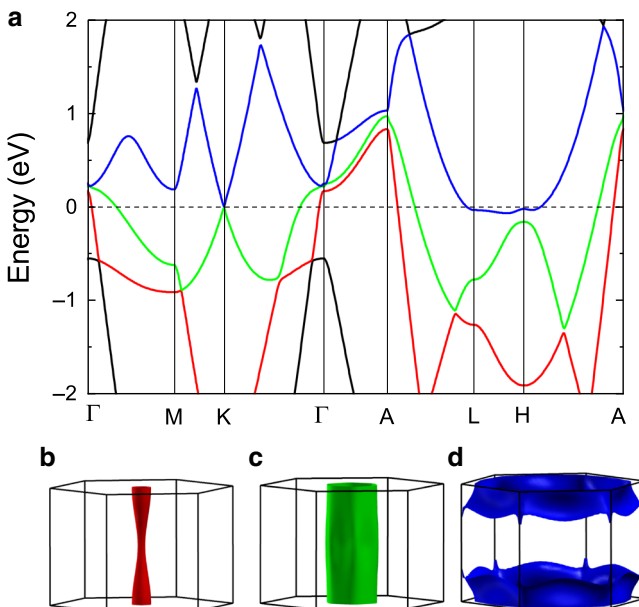

**Fig. 4 Calculated band structure and FS topology of BaGa$_2$. a** Band structure of BaGa$_2$ along high-symmetry paths of the Brillouin zone calculated with SOC. **b** and **c** Two hole-type Fermi surfaces and **d** one electron-type Fermi surface.

**ARPES measurement.** The electronic structure of BaGa$_2$ is measured by ARPES and the quasi-2D Dirac cone at the K point is revealed. Figure 5a shows a schematic drawing of the 3D BZ and its projected surface BZ onto the (00l) surface measured. The FS is shown in Fig. 5b. The most obvious feature is a nearly circular pocket around the $\overline{\text{K}}$ point. Besides this pocket, there are several much weaker features around the $\overline{\Gamma}$ and $\overline{\text{K}}$ points where several bands cross the Fermi energy. This can be seen more clearly in the ARPES spectra along the high-symmetry directions as shown in Fig. 5c. The experimental spectra show an overall agreement with the calculated dispersions at different $k_z$ values. There also exist a few extra band dispersions near the Fermi energy (indicated by red circles in Fig. 5c) originating from surface states which is consistent with the first-principles calculations, as shown in Fig. 5d. Besides, the experimental dispersions which are away from Fermi energy show a mixture of calculated dispersions ranging from $k_z = 0$ and $k_z = 0.5c^*$ (the dispersions at the $k_z$ boundaries are colored black in Fig. 5d), suggesting a very strong $k_z$-broadening[39].

We zoom in the dispersion near $E_F$ to further reveal the Dirac cone at the K point. The dispersion at $k_y = 0$ is shown in Fig. 5f. Due to the matrix element effect[40], the left branch is much weaker than the right branch. To enhance the visualization of the dispersion on both sides, we show in Fig. 5g the normalized spectra by integrating each energy distribution curve (EDC). In addition to the Dirac cone at the K point, there is an extra dispersion originating from the surface state, which is consistent with the calculations. The evolution of Dirac cone along the $k_y$ direction is shown in Fig. 5f. Moving away from the K point, the dispersion shifts down in energy and becomes more parabolic, in agreement with a Dirac cone at the K point as shown in Fig. 5a. The $k_z$-dependence of the Dirac cone is shown in Fig. 5g. Similar behavior is also observed, with the dispersion shifting down in energy when moving away from K point ($k_z = 2c^*$). However, the $k_z$ dispersion of the Dirac cone is much weaker than $k_y$ dispersion and the Fermi velocity along $k_z$ is almost one order of magnitude less than $k_y$, suggesting that the Dirac cone is quasi-2D like centered at K point.

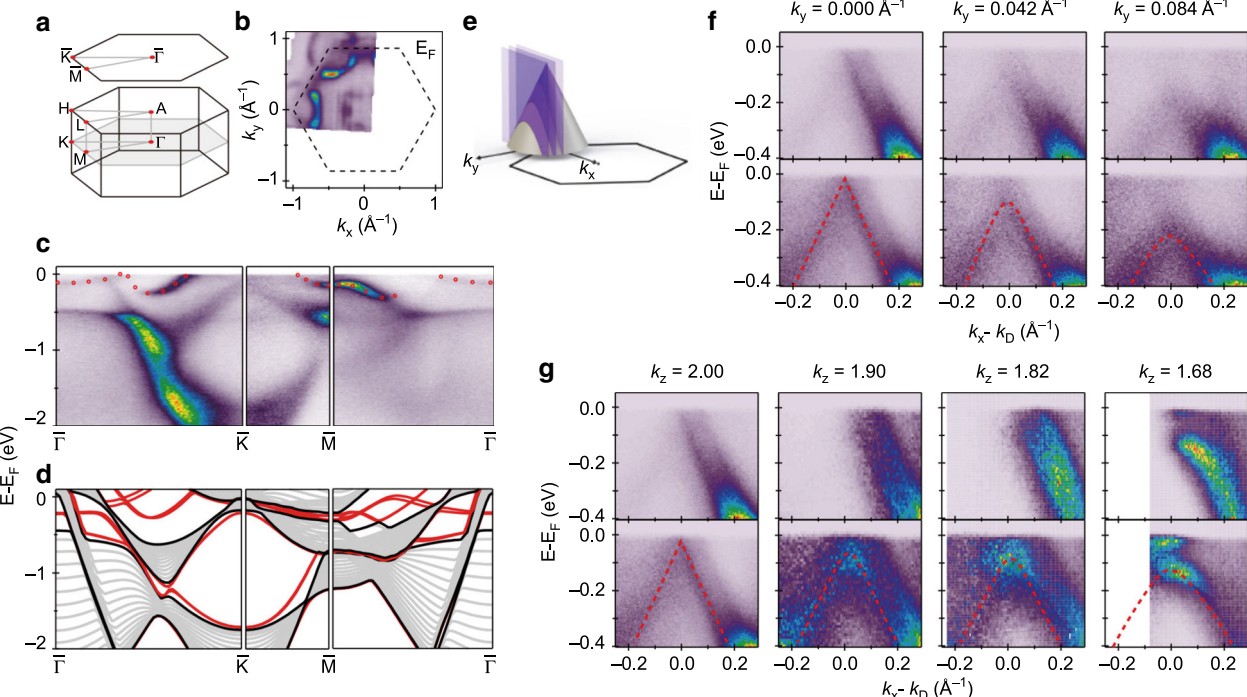

**Fig. 5 Fermi surface, electronic dispersions and quasi two-dimensional Dirac cone in BaGa₂. a** Schematic drawing of the 3D Brillouin zone with high symmetry points marked by red points and its projected surface Brillouin zone to (001) surface. **b** Measured Fermi surface with corresponding Brillouin zone. **c** Measured electronic dispersion along $\overline{\Gamma} - \overline{K} - \overline{M} - \overline{\Gamma}$ direction. **d** Calculated bulk and surface electronic dispersion of the Ba&Ga-terminated BaGa₂ (001) surface with different $k_z$ along $\overline{\Gamma} - \overline{K} - \overline{M} - \overline{\Gamma}$ direction. **e** Schematic drawing shows the slices (purple plane) which cut through the Dirac cones at different $k_y$. **f** Electronic dispersions cut through the Dirac cones at corresponding $k_y$ and corresponding electronic dispersions in the upper panel after normalization by integrating energy distribution curve (EDC). **g** $k_z$ dependence of electronic dispersions cut through the Dirac point and corresponding electronic dispersions in the upper panel after normalization by integrating EDC. The unit of $k_z$ is $2\pi/c$.

**Interlayer transport properties.** BaGa₂ provides an ideal platform to study the unusual interlayer transport properties caused by the tunneling of Dirac fermions between the zeroth LLs since the Ga honeycomb-net layer contributes the quasi-2D Dirac cone and the Ba layer between adjacent two Ga honeycomb-net layers provides the barriers.

Figure 6b presents the angle-dependent interlayer resistivity $\rho_{zz}(\mathbf{B}, \theta)$ at 2.5 K under different fields. It shows the interlayer resistivity peak at $\theta = 90°$. $\theta$ is the angle between **B** and **I** as defined in Fig. 6a. Besides, the interlayer resistivity decreases with the increasing field, resulting in the NIMR at $\theta = 0°$, as shown in Fig. 6b, d and e. The NIMR decreases quickly when temperature changes from 2.5 to 70 K. Both of these anomalous interlayer transport properties can be understood based on the tunneling of Dirac fermions between the zeroth LLs. According to previous works[34,36], the interlayer Dirac fermion tunneling conductivity $\sigma_t^{LL0}(\mathbf{B}, \theta)$ can be described as

$$\sigma_t^{LL0}(\mathbf{B}, \theta) = A|B\cos\theta|\exp\left[-\frac{1}{2}\frac{ec^2(B\sin\theta)^2}{\hbar|B\cos\theta|}\right] \quad (2)$$

where $A$ is a field-independent parameter, $\theta$ is the angle between **B** and **I**, and $c$ is the distance of the adjacent two Ga layers. When the field is applied parallel to **I** along $c$-axis ($\theta = 0°$), the tunneling conductivity is simplified as $\sigma_t^{LL0}(\mathbf{B}, 0°) = A|\mathbf{B}|$. In this case, the tunneling conductivity increases proportional to the magnetic field, leading to the NIMR as exhibited in Fig. 6d. When $\theta$ increases from 0° to 90°, the interlayer tunneling conductivity decreases gradually because of the suppression of 2D LL quantization, resulting in the interlayer resistivity peak as shown in Fig. 6b.

In an ideal 2D Dirac system, the interlayer transport is only contributed by the tunneling of Dirac fermions between the zeroth LLs (tunneling channel) and the resistivity exhibits the equivalent peak values at $\theta = 90°$ under different fields[34,35]. However, as shown in Fig. 6b, the interlayer resistivity peak increases with the increasing field. The inconsistency is reasonable since there also exist trivial bands contributing the positive interlayer MR except for the Dirac band. As discussed in the angle-dependent dHvA oscillations, the results from first-principles calculations, the ratio of $\rho_{xx}(14\,\text{T}, 0°)/\rho_{xx}(14\,\text{T}, 90°) \approx 5$, and $\rho_{zz}/\rho_{xx} \approx 26$, the electronic anisotropy of these trivial bands are analyzed as moderate. Based on these discussion, we assume the trivial bands contribute to the interlayer transport through a momentum relaxation mechanism (i.e. coherent band transport) and describe it with Drude model. Considering both the tunneling mechanism and momentum relaxation mechanism, the total interlayer resistivity is described as

$$\rho_{zz}(\mathbf{B}, \theta) \approx \frac{1}{\sigma_{zz}(\mathbf{B}, \theta)} = \frac{1}{\sigma_t^{LL0}(\mathbf{B}, \theta) + \sigma_c(\mathbf{B}, \theta) + B_0} \quad (3)$$

where $\sigma_c(\mathbf{B}, \theta)$ is the conductivity from the coherent trivial bands. As shown in the Supplementary Table 1, all of the trivial bands do not reach the quantum limit, so the conductivity $\sigma_c(\mathbf{B}, \theta)$ is taken as $\sigma_0/(1 + k \cdot B_{xy}^2)$ for low field approximation ($\sigma_0$ is the Drude conductivity, $k$ is a constant, $B_{xy} = |\mathbf{B}|\sin\theta$). $B_0$ is a fitting parameter[34]. At $\theta = 90°$, the tunneling channel is suppressed [$\sigma_t^{LL0}(\mathbf{B}, 90°) = 0$] and the interlayer resistivity mainly originates from the trivial bands. In such case, the interlayer resistivity is described as $\rho_{zz}(\mathbf{B}, 90°) = 1/\sigma_{zz}(\mathbf{B}, 90°) = 1/(\sigma_c(\mathbf{B}, 90°) + B_0)$. As shown in Fig. 6e the fitting curve matches well with the experimental data. Therefore, to a certain extent, the contribution

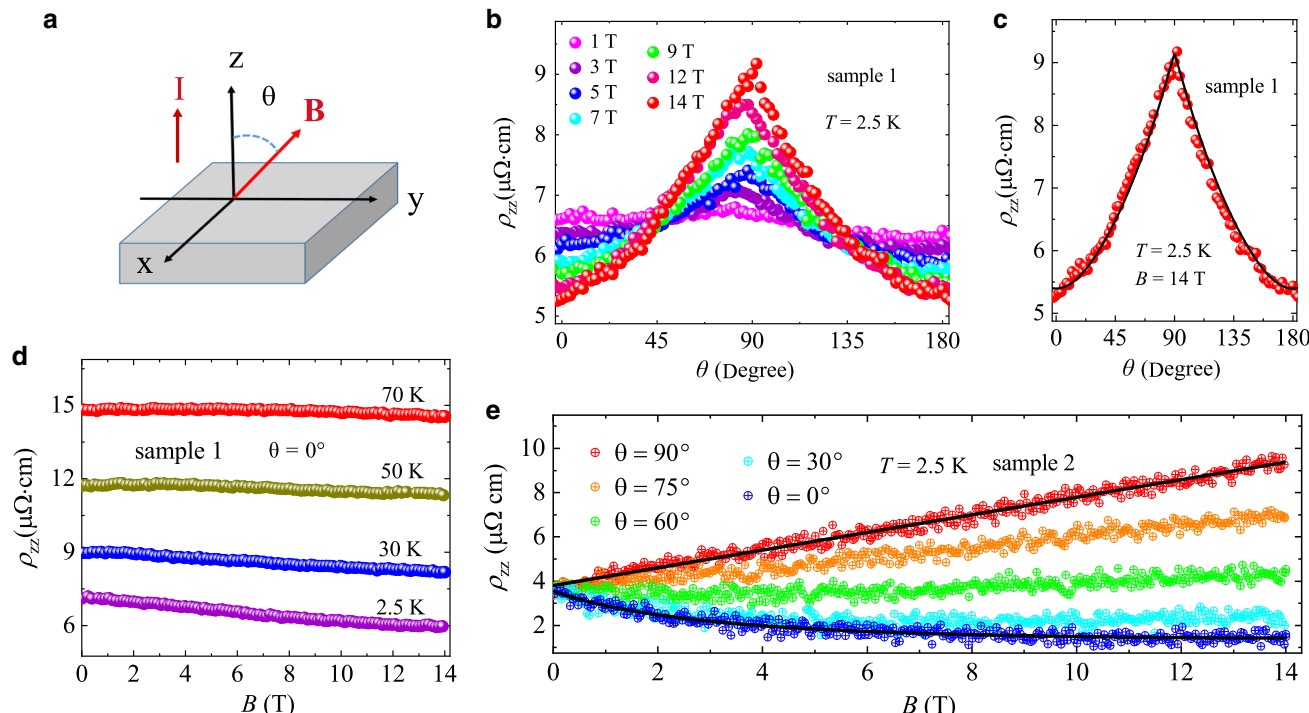

**Fig. 6 Interlayer magnetotransport properties of BaGa$_2$. a** The schematic illustration of interlayer transport experiment setup. **I** flows along $c$, and $\theta$ is the angle between **B** and **I**. **b** Angle dependence of interlayer resistivity $\rho_{zz}$ at 2.5 K under different magnetic fields. **c** Angle dependence of $\rho_{zz}$ at 2.5 K and 14 T (red dots) with the fitting by Eq. (3) (black line). **d** and **e** Field dependence of $\rho_{zz}$ at various temperatures and angles, respectively.

from trivial bands to interlayer transport can be described by Drude model.

The total interlayer transport properties can be considered as the result of the competition between the tunneling mechanism and momentum relaxation mechanism. When $\theta$ is close to 0°, the trivial band-induced positive MR is relatively small because the field-induced scattering is very weak (absence of Lorentz force). However, the density of states (DOS) at Dirac point increases dramatically with the increasing field since the Dirac cone is located exactly at $E_F$. In this case, the tunneling mechanism dominants the interlayer transport, resulting in the NIMR as shown in Fig. 6b, d and e. The angle-dependent and field-dependent unusual interlayer resistivity can be well fitted by Eq. (3), as shown in Fig. 6c and e. When $\theta$ increases from 0° to 90°, the tunneling of Dirac fermions between the zeroth LLs is suppressed while the trivial band induced positive MR increases due to the enhancement of the Lorentz force-induced scattering. It is consistent with the observed interlayer MR, where the NIMR vanishes gradually and positive MR increases with $\theta$ varying from 0° to 90°. These resistivity curves are well fitted by Eq. (3) as shown in Fig. 6c and e. These anomalous interlayer transport properties induced by the tunneling of Dirac fermions between the zeroth LLs have also been observed in (BEDT-TTF)$_2$I$_3$, where the Dirac cone only exists under pressure[34,35]. YbMnBi$_2$ is another quasi-2D topological material that exhibits the same anomalous interlayer transport properties[36], in which the NIMR was not observed possibly due to the large positive MR induced by other bands overwhelming the signature of NIMR. To our knowledge, BaGa$_2$ is the only material that exhibits both NIMR and interlayer resistivity peak at ambient pressure.

## Discussions

To further understand the observed NIMR in BaGa$_2$, we discuss on the possibly different origins of the negative magnetoresistance

(NMR) below. BaGa$_2$ is a non-magnetic material, spin flip may not be a suitable reason for the NIMR in BaGa$_2$[41]. Secondly, the NMR has also been observed in topologically trivial materials at high field such as graphite, where the NMR is induced by the ellipsoidal Fermi surfaces approaching the quantum limit[42]. As displayed in the Supplementary Table 1, all of the trivial bands in BaGa$_2$ are not under the quantum limit with the field of 14 T. In addition, the NIMR in BaGa$_2$ appears once a quite small field is applied. Thus, the scenario of trivial bands under quantum limit does not apply in BaGa$_2$. Thirdly, the NMR induced by chiral anomaly has been widely observed in 3D Dirac and Weyl semimetals[5,9,12,21,22], which usually has the form of $\Delta\sigma_{xx} \propto B^2$ or $B$ depending on the Fermi energy[43]. However, the Dirac cone in BaGa$_2$ opens a small energy gap (~10 meV) when SOC is considered and it will not evolve into Weyl cones when time reversal symmetry is broken (applying the magnetic field). So the chiral anomaly is absent in BaGa$_2$ when **I**//**B**. Besides, the in-plane NMR is not observed when the field is parallel to the current in BaGa$_2$ (Supplementary Fig. 1). Thus the scenario of chiral anomaly induced NMR does not apply either. Fourthly, current jetting can often cause the NMR[44], which has been explained in detail in ref. [45]. Specifically, it is caused by the inhomogeneous spatial distribution of the current in the sample[22,45]. In order to eliminate the current jetting effect, the four-probe AC transport method is applied in the interlayer transport measurement and the current probe with silver paste almost contact fully on the samples' upper and lower surfaces (Supplementary Fig. 2) to make the current flow cross the sample homogeneously. Besides, the NIMR in BaGa$_2$ decreases quickly with the increasing temperature, inconsistent with the current jetting effect[44]. Finally, the NIMR can arise in high-purity layered metal PdCoO$_2$ with the residual resistivity $\rho_{xx}$ ranging from about 10 to 40 nΩ cm in most single crystals[46]. As for BaGa$_2$, the residual resistivity $\rho_{xx}$ and $\rho_{zz}$ are about 440 nΩ cm and 11.44 μΩ cmwhich are larger than that of PdCoO$_2$ at least one order of magnitude. So the condition $4t_c >$

$\hbar\omega_c$ $(\varepsilon_n = (n + 1/2)\hbar\omega_c + 2t_c\cos(k_z c)$, $t_c$ is interlayer transfer integral) describing the Landau levels for quasi-2D Fermi surface under the magnetic field in the quasi-2D metal material $PdCoO_2$ may not be fulfilled here. Besides, the interlayer MR in $PdCoO_2$ increases quickly at low field and decreases at high field when field deviates from $c$-axis which is quite different from that of $BaGa_2$. From the above, we conclude that the interlayer NMR and the interlayer resistivity peak result from the tunneling of Dirac fermions between the zeroth LLs.

In summary, $BaGa_2$ is a multilayered Dirac semimetal with quasi-2D Dirac cone located at $E_F$, which is confirmed by the first-principles calculations, angle-dependent dHvA oscillations and ARPES measurements. The unusual interlayer transport properties (NIMR and peak in angle-dependent interlayer resistivity) originating from the tunneling of Dirac fermions on the zeroth LLs are both observed in $BaGa_2$ under ambient pressure for the first time. Our findings highlight the unusual role of the zeroth LLs in interlayer magnetotransport and enrich the novel transport properties of topological materials at quantum limit.

## Methods

**Crystal growth and magnetotransport measurements.** The single crystal $BaGa_2$ was grown by self-flux method. The ingredient ratio with Ba:Ga = 41.7:58.3 were put into the alumina crucible and sealed into a quartz ampoule. Then the ampoule was heated to 950 °C and kept for 50 h to make sure the materials have melted and mixed thoroughly. The process of crystal growth was carried up in cooling at 3 °C/h to 700 °C. The $BaGa_2$ single crystal was obtained after centrifuging the excess flux at that temperature. The atomic composition of $BaGa_2$ was checked by energy-dispersive x-ray spectroscopy (EDS, Oxford X-Max 50). The crystal structure was determined by XRD in Bruker D8 Advance x-ray diffractometer. TOPAS-4.2 was employed for refinement. The electric transport and magnetic properties were collected in Quantum Design PPMS-14 T and MPMS-3 SQUID VSM system.

**Angle-resolved photoemission spectroscopy.** ARPES measurements were taken at BL13U of Hefei National Synchrotron Radiation Laboratory and our home laboratory. The crystals were cleaved in situ and measured at a temperature of $T \approx$ 15 K in vacuum with a base pressure better than $1 \times 10^{-10}$ torr.

**Band structure calculations.** The electronic structures of $BaGa_2$ were studied by using first-principles calculations. The projector augmented wave (PAW) method[47,48] as implemented in the VASP package[49–51] was used to describe the core electrons. The SCAN type exchange-correlation potential was adopted[52]. The kinetic energy cutoff of the plane-wave basis was set to be 400 eV. A $20 \times 20 \times 18$ $k$-point mesh was utilized for the BZ sampling and the Fermi surface was broadened by the Gaussian smearing method with a width of 0.05 eV. Both cell parameters and internal atomic positions were fully relaxed until the forces on all atoms were smaller than 0.01 eV/A. Once the equilibrium structures were obtained, the electronic structures were calculated with the SOC effect. The Fermi surfaces were studied by using the maximally localized Wannier functions (MLWF) method[53,54]. To study the surface states, a $1 \times 1$ two-dimensional supercell with an $BaGa_2$ slab containing 59 atoms and a 20 Å vacuum was employed to simulate the $BaGa_2$ (001) surface.

## Data availability

The authors declare that the data supporting the findings of this study are available within the article and its Supplementary Information. Extra data are available from the corresponding authors upon reasonable request.

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

## Acknowledgements

We thank T. Qian, G.F. Chen, and H.C. Lei for helpful discussions. This work is supported by the National Natural Science Foundation of China (Nos.11874422, 11574391, 11725418, 11774422, 11774424), the Ministry of Science and Technology of China (Nos. 2019YFA0308602, 2019YFA0308603, 2017YFA0302903) and the Fundamental Research Funds for the Central Universities, and the Research Funds of Renmin University of China (Nos. 18XNLG14, 19XNLG18, 19XNLG13). Computational resources were provided by the Physical Laboratory of High Performance Computing at Renmin University of China. The Fermi surfaces were prepared with the XCRYSDEN program[55].

## Author contributions

T.-L.X. coordinated the project and designed the experiments. S.X. and Y.-Y.W. synthesized the single crystals of $BaGa_2$ with the assistance of Q.-H.Y., L.-L.S. and Y.S. C.B. and S.Z. performed the ARPES measurements. P.-J.G., K.L. and Z.-Y.L. performed ab initio calculations. S.X. performed the magneto-transport measurements with the assistance of Y.-Y.W. S.X., C.B. and P.-J.G. plotted the figures and analyzed the experimental data. S.X., Y.-Y.W., and T.-L.X. wrote the paper. All authors discussed the results and commented on the manuscript.

## Competing interests

The authors declare no competing interests.
