## [Peer Review File · Nature Communications]

Reviewers' comments:

Reviewer #1 (Remarks to the Author):

This paper reports the observation of negative interlayer magnetoresistance (NIMR) in a novel Dirac semimetal BaGa₂. The author synthesized the BaGa₂ crystal, which had been predicted as a Dirac semimetal, and they thoroughly performed the characterization of its electronic structure using dHvA oscillations, ARPES, and DFT calculation. They confirmed that there exist one hole doped Dirac cones at K-point, two cylindrical hole Fermi surfaces, and one electron Fermi surfaces. Then, they investigated interlayer magnetotransport, and found the NIMR, and its characteristic field-orientation-dependence with a peak at the field direction parallel to layers. The authors explained the NIMR by the interlayer tunneling between the zeroth Landau levels (LLs) characteristic to the layered Dirac fermion systems such as an organic conductor a-(BEDT-TTF)₂I₃.

This is the first report of electronic structure and properties of BaGa₂. Therefore, it is basically worth for publishing. However, there seems a few unclear points or contradictions about the origin of the NIMR, which seems the main issue of this paper.

(1) BaGa₂ has a few Fermi surfaces in addition to the hole-doped Dirac cone at the K-point. I think that these majority carriers must dominantly contribute to the interlayer magnetoconductivity. Their contribution, that is, $\sigma_c(B, \theta)$ in Eq. (2), is not necessarily given by the Drude form because the Fermi surfaces are open. Moreover, the pre-factor σ_0 is not qualitatively evaluated in the text. The author had better clarify the absolute value (zero level) of interlayer resistivity in Fig. 6(b)(c)(d). I think that the contribution of tunneling of Dirac fermions is small. In fact, in the Dirac/Weyl systems coexisting with the trivial carriers, the negative magnetoresistance due to chiral anomaly is very small [for example, Nat. Mat. 16, 1090 (2017)].

(2) According to FIG. 4, the green and blue bands touch at the K-point forming a 3D Dirac cone (when SOC is ignored). Therefore, BaGa₂ is a 3D Dirac semimetal, and not a nodal-line Dirac semimetal such as a-(BEDT-TTF)₂I₃. In this case, the LLs has k_z-dispersion and the zeroth LL becomes a chiral LL (subband) characteristic to Weyl/Dirac semimetals. Since the degeneracy of the chiral LL is proportional to B_z, we can similarly expect the NIMR at the quantum limit. However, this is nothing but the chiral anomaly at the quantum limit. In addition, the tunneling picture of layered Dirac fermion system can be applied only for the quasi-2D conductors with incoherent interlayer coupling, in which the interlayer dispersion (interlayer tunneling probability) is smaller than the scattering broadening (in-plane scattering probability). However, BaGa₂ seems a highly-anisotropic 3D conductor with the well-defined k_z-dispersion (interlayer coherence).

The followings are minor technical mistakes.

(3) The horizontal axis of FIG. 2(b) is not " $B(T^{-1})$ " but " $1/B(T^{-1})$ ".

(4) page 4, right column, 6 lines from the bottom: "(BEDT-FFT)₂I₃" -> "(BEDT-TTF)₂I₃"

In summary, I think that this paper is basically worth for publication, but some revision is necessary for discussing the physical origin of NIMR.

Reviewer #2 (Remarks to the Author):

Xu et al studied the electronic structure and the magneto-transport of BaGa₂ in theory and experiment thoroughly. They found negative magnetoresistance when a magnetic field is parallel with the current which is claimed as the negative interlayer magnetoresistance. Their results are quite interesting and well established. The negative magneto-resistance in the Dirac semimetal is also quite interesting in the community. Before my decision, I would like the authors to consider the following questions.

1. The negative magnetoresistance could come from the current jetting effect. It should be discussed

in the paper when mentioning the negative magnetoresistance. I would ask the authors to discuss how do the eliminate this effect.

2. I suggest the authors to calculate the dHvA frequencies from the first-principle calculations since It's straight forward when they have the Fermi surface.

3. The authors mentioned the pocket at the K point in the ARPES measurement. However, they didn't do the DFT simulations to check whether it's the surface states. It would be better if the author can do a DFT simulation to check that. First, we can a finite thick slab system like 10 unit cells. Then do a structure relaxation. In the end, calculate the band structure.

Reviewer #1 (Remarks to the Author):

This paper reports the observation of negative interlayer magnetoresistance (NIMR) in a novel Dirac semimetal BaGa₂. The author synthesized the BaGa₂ crystal, which had been predicted as a Dirac semimetal, and they thoroughly performed the characterization of its electronic structure using dHvA oscillations, ARPES, and DFT calculation. They confirmed that there exist one hole doped Dirac cones at K-point, two cylindrical hole Fermi surfaces, and one electron Fermi surfaces. Then, they investigated interlayer magnetotransport, and found the NIMR, and its characteristic field-orientation-dependence with a peak at the field direction parallel to layers. The authors explained the NIMR by the interlayer tunneling between the zeroth Landau levels (LLs) characteristic to the layered Dirac fermion systems such as an organic conductor α -(BEDT-TTF)₂I₃.

This is the first report of electronic structure and properties of BaGa₂. Therefore, it is basically worth for publishing. However, there seems a few unclear points or contradictions about the origin of the NIMR, which seems the main issue of this paper.

Reply: We thank the referee for reviewing our manuscript and we are also grateful to the referee since he/she has seen the potential impact of our work. We apologize for not claiming clearly about the origin of negative interlayer magnetoresistance (NIMR) in the original manuscript and appreciate the referee's insightful and valuable comments. We have now addressed all of the points raised by the referee and revised our manuscript accordingly. Here, we present the point-to-point reply to the comments/suggestions.

(1) BaGa₂ has a few Fermi surfaces in addition to the hole-doped Dirac cone at the K-point. I think that these majority carriers must dominantly contribute to the interlayer magnetoconductivity. Their contribution, that is,

sigma_c(B,theta) in Eq.(2), is not necessarily given by the Drude form because the Fermi surfaces are open. Moreover, the pre-factor sigma_0 is not qualitatively evaluated in the text. The author had better clarify the absolute value (zero level) of interlayer resistivity in Fig.6(b)(c)(d). I think that the contribution of tunneling of Dirac fermions is small. In fact, in the Dirac/Weyl systems coexisting with the trivial carriers, the negative magnetoresistance due to chiral anomaly is very small [for example, Nat. Mat. 16, 1090 (2017)].

BaGa2 has a few Fermi surfaces in addition to the hole-doped Dirac cone at the K-point. I think that these majority carriers must dominantly contribute to the interlayer magnetoconductivity.

Reply: We thank the referee again for his/her insightful comments. In addition to the Dirac cone at K point, there do exist three trivial Fermi pockets in BaGa₂. All of these pockets contribute to the interlayer transport. We have considered all of these contributions in our analysis of the interlayer magnetoresistance (MR) data in Figs. 6(c) and 6(e), as shown in the equation, *i.e.*

$$\rho_{zz}(B,\theta) \approx \frac{1}{\sigma_{zz}(B,\theta)} = \frac{1}{\sigma_t^{LL0}(B,\theta) + \sigma_c(B,\theta) + B_0} \quad (1)$$

where $\sigma_c(B,\theta)$ is the conductivity originating from the trivial pockets by the momentum relaxation process, σ_t^{LL0} represents the tunneling conductivity of Dirac fermions, and B_0 is a fitting parameter [1].

When B is vertical to the current ($\theta=90^\circ$), the tunneling of Dirac fermion vanishes and $\sigma_t^{LL0}(B,90^\circ)=0$ because of the suppression of 2D LL quantization. In this case, the interlayer transport is mainly contributed by the trivial bands. In an ideal 2D Dirac system, the interlayer transport is only contributed by the tunneling of Dirac fermions between the zeroth LLs (tunneling channel) and the resistivity exhibits the equivalent peak values at $\theta=90^\circ$ under different fields [1,2]. In our case, as shown in the revised manuscript Fig. 6(b), the interlayer

resistivity peak of BaGa₂ increases with the increasing field at $\theta=90^\circ$, which is inconsistent with the ideal 2D Dirac system. The inconsistency is reasonable since there exist trivial bands except for the Dirac points in BaGa₂, which contribute the positive interlayer MR. The total interlayer transport properties can be considered as the result of the competition between the tunneling mechanism and the momentum relaxation mechanism.

When θ is close to 0° , the trivial band-induced positive MR is relatively small because the field-induced scattering is very weak (absence of Lorentz force). However, the density of states (DOS) at Dirac point increase dramatically with the increasing field since the Dirac cone is located exactly at E_F . In this case, the tunneling mechanism dominates the interlayer transport, resulting in the negative interlayer MR as shown in Figs.6(b) and 6(d)-6(e).

The field-dependent unusual interlayer resistivity is well fitted by Eq. (1), as shown in Fig. 6(e) [Reply Fig. 1(e)]. When θ increases from 0° to 90° , the tunneling of Dirac fermion between the zeroth LLs is suppressed while the trivial band-induced positive MR increases due to the enhancement of the Lorentz force-induced scattering. It is consistent with the observed interlayer MR, where the NIMR vanishes gradually and positive MR increases with θ varying from 0° to 90° . These resistivity curves are well fitted by Eq. (1) as shown in Figs. 6(c) and 6(e) [Reply Figs. 1(c) and 1(e)].

Their contribution, that is, $\sigma_c(B,\theta)$ in Eq.(2), is not necessarily given by the Drude form because the Fermi surfaces are open.

Reply: We apologize for not making our claims clear enough to the reviewer in the original manuscript and we appreciate the reviewer's insightful comments, which have been very helpful in improving the quality of our manuscript. As shown in the Figs. 4(b) and 4(c), the hole-like open Fermi surfaces in BaGa₂ with lotus-root shape indicate that there exists k_z -dispersion. The electron-like Fermi surface also exhibits well-defined k_z -dispersion. It is consistent with the

observed moderate electronic anisotropy at 2.5 K ($\rho_{zz} / \rho_{xx} \approx 26$) and magneto-transport properties at 2.5 K ($\rho_{xx}(14T, 0^\circ) / \rho_{xx}(14T, 90^\circ) \approx 5$, the current is always perpendicular to the field), as shown in Figs. 1(e) and 3(d) in the manuscript. If the trivial bands were highly 2D-like, a much larger electronic anisotropy would be expected, which was not observed in the experiments. Thus, we assume the trivial bands contribute to the interlayer transport through a momentum relaxation mechanism (i.e. coherent band transport) and describe it with Drude model. In addition, the tunneling channel is suppressed when $B \perp I$ and the interlayer resistivity mainly originates from the trivial bands. In such case, the interlayer resistivity is described by Drude model as $\rho_{zz}(B, \theta) \approx \frac{1}{\sigma_{zz}(B, \theta)} = \frac{1}{\sigma_c(B, \theta)}$. As shown in the revised manuscript Fig. 6(e), the fitting curve matches well the experimental data when $B \perp I$ ($\theta=90^\circ$). Therefore, to a certain extent, the contribution from trivial bands to interlayer transport can be described by Drude model.

Moreover, the pre-factor σ_0 is not qualitatively evaluated in the text. The author had better clarify the absolute value (zero level) of interlayer resistivity in Fig. 6(b)(c)(d).

Reply: We thank the referee for this valuable suggestion. The Fig. 6 has been revised in the manuscript to better illustrate the unusual interlayer transport characteristics. And the absolute values of the interlayer resistivity in Figs. 6 are clarified accordingly in the revised manuscript. The revised Fig. 6 is shown below.

Reply Fig.1 (Fig. 6 in the revised manuscript) (a) The interlayer transport experiment setup, and θ is the angle between B and I . (b) Angle dependence of interlayer resistivity ρ_{zz} at 2.5 K and different magnetic fields. (c) Angle dependence of ρ_{zz} at 2.5 K and 14 T (dots) and the fits by Eq. (3) (line). (d)(e) Field dependence of ρ_{zz} at various temperatures and sample positions, respectively.

I think that the contribution of tunneling of Dirac fermions is small. In fact, in the Dirac/Weyl systems coexisting with the trivial carriers, the negative magnetoresistance due to chiral anomaly is very small [for example, Nat. Mat. 16, 1090 (2017)].

Reply: We totally agree with the referee that the negative MR due to chiral anomaly is very small in Dirac/Weyl systems with the trivial carriers. α -(BEDT-TTF)₂I₃ is an ideal quasi-2D Dirac system which only possesses the quasi-2D Dirac bands under pressure. Its interlayer transport only originates from interlayer tunneling, exhibiting a large NIMR [1,2]. However, the NIMR is absent in YbMnBi₂ because its interlayer transport originates from both the tunneling of quasi-2D Weyl fermion between the zeroth LLs and Dirac bands (far away from quantum limit) [3]. Its NIMR signature is overwhelmed by the Dirac bands induced large positive MR.

In addition to the quasi-2D Dirac cone at K point, BaGa₂ also possesses two

lotus-root shape hole-like FSs and one electron-like FS, which exhibit a moderate electronic anisotropy $\rho_{zz} / \rho_{xx} \approx 26$ at zero field. The contribution to the interlayer transport from these trivial bands are relatively small because of the anisotropic electronic structure. However, there are 6 (in fact, it should be 2 considering the extended Brillouin zones) quasi-2D Dirac cones in the first Brillouin zone in BaGa₂, and the DOS (Dirac fermion) at E_F increases dramatically when the magnetic field is applied. The tunneling mechanism dominates the interlayer transport, resulting in the NIMR.

(2) According to FIG.4, the green and blue bands touch at the K-point forming a 3D Dirac cone (when SOC is ignored). Therefore, BaGa₂ is a 3D Dirac semimetal, and not a nodal-line Dirac semimetal such as a-(BEDT-TTF)₂I₃. In this case, the LLs has k_z -dispersion and the zeroth LL becomes a chiral LL (subband) characteristic to Weyl/Dirac semimetals. Since the degeneracy of the chiral LL is proportional to B_z , we can similarly expect the NIMR at the quantum limit. However, this is nothing but the chiral anomaly at the quantum limit. In addition, the tunneling picture of layered Dirac fermion system can be applied only for the quasi-2D conductors with incoherent interlayer coupling, in which the interlayer dispersion (interlayer tunneling probability) is smaller than the scattering broadening (in-plane scattering probability). However, BaGa₂ seems a highly-anisotropic 3D conductor with the well-defined k_z -dispersion (interlayer coherence).

According to FIG.4, the green and blue bands touch at the K-point forming a 3D Dirac cone (when SOC is ignored). Therefore, BaGa₂ is a 3D Dirac semimetal, and not a nodal-line Dirac semimetal such as a-(BEDT-TTF)₂I₃. In this case, the LLs has k_z -dispersion and the zeroth LL becomes a chiral LL (subband) characteristic to Weyl/Dirac semimetals. Since the degeneracy of the chiral LL is proportional to B_z , we can similarly expect the NIMR at the quantum limit.

However, this is nothing but the chiral anomaly at the quantum limit.

Reply: We thank the referee again for his/her insightful and valuable comments. Clarification of this issue is crucial to the understanding of our claim. As pointed out by the referee, the LL energy depends on the momentum k_z along the field direction in a 3D Dirac system which is given by $\varepsilon_n = \pm v_F \sqrt{2ehB|n| + k_z^2}$ ($n=0, \pm 1, \pm 2, \dots$) [4]. In 2D Dirac system, the LL energy is described as $\varepsilon_n = \pm v_F \sqrt{2ehB|n|}$ ($n=0, \pm 1, \pm 2, \dots$) [5]. Under this circumstance, the zeroth LL is always locked at the Dirac node and there exist no k_z -dispersion. Our ARPES results and band structure studies strongly suggest that the Dirac band in BaGa₂ are much quasi-2D-like and the Dirac cone is highly anisotropic since its Fermi velocity V_F along the k_z direction is almost one order of magnitude less than V_F along the k_y direction, as shown in the manuscript Figs. 5(f) and 5(g). Figures. 5(f) and 5(g) also indicate that the Dirac point is located exactly at the E_F . These characteristics of Dirac cone in BaGa₂ have also been revealed in previous work [6]. Thus, the zeroth LLs of Dirac fermion in BaGa₂ are always locked at E_F when the magnetic field is applied. The DOS (Dirac fermions) at the zeroth LLs increases dramatically with the increasing field, leading to the tunneling of the zeroth LLs' Dirac fermion between the adjacent two honeycomb Ga layers.

In fact, the Dirac cone in BaGa₂ opens a small energy gap (less than 10 meV) when SOC is considered and the Dirac cone will not evolve into Weyl cone when time reversal symmetry is broken (applying the magnetic field). So the chiral anomaly is absent in BaGa₂ when $I//B$. We didn't observe the in-plane longitudinal negative magnetoresistance with $B//I$, as shown in the Supplementary Fig. 1. Although the Dirac cone at K points opens a very little gap when considering SOC, the low-energy excitation can still be described by the Dirac equation and possesses the characteristic of Dirac fermions.

In addition, the tunneling picture of layered Dirac fermion system can be applied

only for the quasi-2D conductors with incoherent interlayer coupling, in which the interlayer dispersion (interlayer tunneling probability) is smaller than the scattering broadening (in-plane scattering probability). However, BaGa₂ seems a highly-anisotropic 3D conductor with the well-defined k_z-dispersion (interlayer coherence).

Reply: We totally agree with the referee that the tunneling picture of layered Dirac fermion system can be applied only for the quasi-2D conductors with incoherent interlayer coupling. The crystal structure of BaGa₂ consists of Ba layers and Ga honeycomb-net layers as shown in Figs. 1(a) and 1(c). According to our first-principles calculations and previous work [6], the quasi-2D Dirac cone originates from the Ga p_z orbitals, which is confirmed by our ARPES measurement as the Fermi velocity V_F along the k_z direction being almost one order of magnitude less than V_F along the k_y direction. The Ba layers between the adjacent Ga honeycomb-net layer act as barriers for the interlayer transport. Hence, the tunneling picture between the adjacent two Ga honeycomb-net layers is reasonable.

In addition to the quasi-2D Dirac cone, there are also trivial bands with well defined k_z-dispersion. We assume the contribution from the trivial bands can be described by momentum relaxation mechanism (i.e. coherent band transport).

The total interlayer resistivity is described as

$$\rho_{zz}(B, \theta) \approx \frac{1}{\sigma_{zz}(B, \theta)} = \frac{1}{\sigma_t^{LL0}(B, \theta) + \sigma_c(B, \theta) + B_0},$$

which consists of tunneling conductivity $\sigma_t^{LL0}(B, \theta)$ of Dirac fermions between the zeroth LLs (interlayer incoherent), conductivity from trivial bands $\sigma_c(B, \theta)$ (interlayer coherent) and fitting parameter B₀. The interlayer resistivity can be considered as the result of the competition between the tunneling mechanism (interlayer incoherent) and the momentum relaxation process of trivial pockets (interlayer coherent). The negative interlayer magnetoresistance vanishes gradually with B rotating from

$B//I$ ($\theta=0^\circ$) to $B \perp I$ ($\theta=90^\circ$) as shown in the manuscript Fig. 6(e), indicating that the interlayer resistivity is dominated by the tunneling mechanism when θ is close to 0° and by the momentum relaxation mechanism when θ is close to 90° .

The followings are minor technical mistakes.

(3) The horizontal axis of FIG.2(b) is not " $B(T^{-1})$ " but " $1/B(T^{-1})$ ".

(4) page 4, right column, 6 lines from the bottom: "(BEDT-FFT)2I3"->"(BEDT-TTF)2I3"

Reply: The typo in the manuscript is reexamined and revised accordingly.

In summary, I think that this paper is basically worth for publication, but some revision is necessary for discussing the physical origin of NIMR.

Reply: We thank the referee again to review our manuscript and give us such important suggestions to improve our manuscript. We have addressed all the points raised by the referee and revised the manuscript accordingly.

Reference:

- [1] T. Osada, J. Phys. Soc. Jpn. **77**, 084711 (2008).
- [2] N. Tajima, S. Sugawara, R. Kato, Y. Nishio, and K. Kajita, Phys. Rev. Lett. **102**, 176403 (2009).
- [3] J. Liu, J. Hu, D. Graf, T. Zou, M. Zhu, Y. Shi, S. Che, S. Radmanesh, C. Lau, L. Spinu, *et al.*, Nat. Commun. **8**, 646 (2017).
- [4] B. A. Bernevig & L. H. Taylor, Topological Insulators and Topological Superconductors. (Princeton University Press, 2013).
- [5] T. Ando, Prog. Theor. Phys. Suppl. **176**, 203 (2008).
- [6] Q. Gibson, L. Schoop, L. Muechler, L. Xie, M. Hirschberger, N. Ong, R. Car, and R. Cava, Phys. Rev. B **91**, 205128 (2015).

Reviewer #2 (Remarks to the Author):

Xu et al studied the electronic structure and the magneto-transport of BaGa₂ in theory and experiment thoroughly. They found negative magnetoresistance when a magnetic field is parallel with the current which is claimed as the negative interlayer magnetoresistance. Their results are quite interesting and well established. The negative magneto-resistance in the Dirac semimetal is also quite interesting in the community. Before my decision, I would like the authors to consider the following questions.

Reply: We thank the referee for reviewing our manuscript. We are also grateful to the referee that he/she has seen the potential impact of our work. We appreciate for his/her insightful and valuable comments/suggestions which have been very helpful in further improving our manuscript. In the following, we present the point-to-point response.

1. The negative magnetoresistance could come from the current jetting effect. It should be discussed in the paper when mentioning the negative magnetoresistance. I would ask the authors to discuss how do the eliminate this effect.

Reply: We thank the referee again for his/her insightful and valuable comments. Clarification of this issue is crucial to the understanding of our claim. A negative longitudinal MR can arise from current jetting effect, which has been explained in detail in Ref. [1]. Specifically, it is caused by the inhomogeneous spatial distribution of the current in the sample [1, 2]. In order to eliminate the current jetting effect, the four-probe AC transport method is adopted in the interlayer transport measurement and the probes of current with silver paste contact almost completely on the samples' upper or lower surfaces [Supplementary Fig. 2(a)] to make sure that the current flows cross the sample homogeneously. Besides, the observed negative interlayer MR in BaGa₂ decreases dramatically with the increasing temperature as shown in the Reply

Fig. 1(d). By contrast, the negative MR originating from current jetting does not exhibit the monotonic dependence on the temperature [3]. To further confirm the conclusion, we also carried out transport measurements on other samples shaped into rectangle with four silver paste contacts fully crossing their width [Supplementary Fig. 2(b)].

[Redacted]

Reply Fig. 1 (a)(b)(c) Current jetting induced negative MR in $\text{Ag}_{2+\delta}\text{Te}$, $\text{Ag}_{2-\delta}\text{Te}$ and $\text{Ag}_{2+\delta}\text{Se}$, respectively, which exhibit temperature independent characteristic [3]. (d) Negative interlayer negative MR in BaGa2 at different temperatures.

2. I suggest the authors to calculate the dHvA frequencies from the first-principle calculations since It's straight forward when they have the Fermi surface.

Reply: We thank the referee for his/her insightful and valuable suggestions. According to the Onsager relation $F = (\Phi_0 / 2\pi^2) A = (\hbar / 2\pi e) A$, the frequency F in the dHvA oscillations is proportional to the extreme cross sectional area A of the Fermi surface normal to the magnetic field. According to the first-principles calculations in our original manuscript, there is a hole-doped Dirac cone at K point. Theoretically, this hole-type Fermi surface should result in a frequency in

the dHvA oscillations as the mobility from the Dirac band is very high. However, our ARPES results reveal that the Dirac point locates exactly at E_F . In the revised manuscript we improved the calculation method (The SCAN type exchange-correlation potential was adopted) and found the Dirac point does locate at E_F (Reply Fig. 2), which is consistent with previous work [4]. The calculated Fermi surfaces with different calculation methods show slight difference. In addition to the Dirac cone, our first-principles calculations reveal that there is one electron-type pocket along L-H and two hole-type pockets at Γ point in the first Brillouin zone. However, according to previous work, there seems to be an analogous hole-type pocket at H point and only one hole-type pocket at Γ point [4]. Besides, in real materials the Fermi surface can be affected by many factors such as temperature, magnetic field, or defects (doping) *etc.* According to our calculations, the calculated dHvA frequencies from the hole-type Fermi surface [Reply Fig. 2(b)] are 61.4 T and 430.1 T corresponding to the observed frequencies F_β (56.6 T) and F_γ (356 T). The calculated frequency from the hole-type Fermi surface [Reply Fig. 2(c)] is 2273.4 T corresponding to the observed frequency F_η (1862 T). The slight difference may arise from above mentioned origins. The calculated largest frequency from the electron-type Fermi surface [Reply Fig. 2(d)] is 4608.3 T which is too high to be observed in the experiments. The observed frequency F_α may originate from the electron-type Fermi surface (Reply Fig. 2(d)) as there is a branch along K-H and an analogous hole-type pocket at H point[4].

Reply Fig. 2. (a) Band structure of BaGa₂ along high symmetry lines of the Brillouin zone calculated with SOC. (b)(c) Two hole-type Fermi surfaces and (d) one electron-type Fermi surface of BaGa₂.

3. The authors mentioned the pocket at the K point in the ARPES measurement. However, they didn't do the DFT simulations to check whether it's the surface states. It would be better if the author can do a DFT simulation to check that. First, we can a finite thick slab system like 10 unit cells. Then do a structure relaxation. In the end, calculate the band structure.

Reply: We thank the referee for this insightful and valuable suggestion. To check whether the pocket at the K point in the ARPES measurement is the surface states, a 1×1 two-dimensional supercell with a 20 unit cells BaGa₂ slab and a 20 Å vacuum was employed to simulate the Ba&Ga-terminated BaGa₂ (001) surface [Reply Fig. 2(a)] and Ga-terminated BaGa₂ (001) surface [Reply Fig. 2(b)]. The calculated results indicate that there do exist the surface states around the K point (Fig. 2). Thus the pocket at the K point in the ARPES measurement is indeed from the surface states.

Reply Fig. 2. Surface band structure of the Ba&Ga-terminated (a) and Ga-terminated (b) BaGa₂ (001) surface along the high-symmetry directions of the surface Brillouin zone.

Reference:

- [1] A. B. Pippard, *Magnetoresistance in metals*, vol. 2 (Cambridge University Press, 1989).
- [2] C.-L. Zhang, S.-Y. Xu, I. Belopolski, Z. Yuan, Z. Lin, B. Tong, G. Bian, N. Alidoust, C.-C. Lee, S.-M. Huang, *et al.*, *Nat. Commun.* **7**, 10735 (2016).
- [3] J. Hu, T. Rosenbaum, and J. Betts, *Phys. Rev. Lett.* **95**, 186603 (2005).
- [4] Q. Gibson, L. Schoop, L. Muechler, L. Xie, M. Hirschberger, N. Ong, R. Car, and R. Cava, *Phys. Rev. B* **91**, 205128 (2015).

REVIEWERS' COMMENTS:

Reviewer #1 (Remarks to the Author):

This paper reports the observation of negative interlayer magnetoresistance (NIMR) in a novel Dirac semimetal BaGa₂. First, the authors confirmed that there exist one hole doped Dirac cones at K-point using quantum oscillations, ARPES, and DFT calculation. Then, they observed the NIMR, and explained it by the interlayer tunneling between the zero-th Landau levels (LLs) characteristic to the multilayer Dirac fermion system.

This paper is the first report of electronic structure and properties of BaGa₂, and most of unclear points for me have been clarified in the new manuscript. Therefore, I think that this paper is worth for publication.

Reviewer #2 (Remarks to the Author):

The authors well answered the raised questions. They improved manuscript is suitable to be published in Nature communication. I recommend it to be published in this journal.

Reviewer #4 (Remarks to the Author):

Xu et al. reported negative interlayer magneto-resistance (NIMR) in BaGa₂ and interpreted their observation by the model of tunneling between Dirac fermions in the quantum limit. The technical part of the paper is well-written. The authors described their data and analyses clearly. I have also read the replies to the review report. I appreciate the authors' careful replies. Therefore, I would recommend the paper for publication. I have the following suggestions/comments.

1. It seems that the authors started to talk about the NIMR and "tunneling between Dirac fermions in the quantum limit" from the very beginning by citing previous works. No qualitative physical pictures were given. Given the broad audience of the Nature journals, would encourage the authors to make some schematics in a figure and also have a few sentences describing the physical picture.
2. While the technical descriptions are clear, I didn't get from reading the manuscript why observing the NIMR and "tunneling between Dirac fermions in the quantum limit" would be novel or significant. I would encourage the authors to do a better job in terms of convey the novelty or significance of their results.

Reviewer #1 (Remarks to the Author):

This paper reports the observation of negative interlayer magnetoresistance (NIMR) in a novel Dirac semimetal BaGa₂. First, the authors confirmed that there exist one hole doped Dirac cones at K-point using quantum oscillations, ARPES, and DFT calculation. Then, they observed the NIMR, and explained it by the interlayer tunneling between the zero-th Landau levels (LLs) characteristic to the multilayer Dirac fermion system.

This paper is the first report of electronic structure and properties of BaGa₂, and most of unclear points for me have been clarified in the new manuscript. Therefore, I think that this paper is worth for publication.

Reply: We thank the referee for his/her recommendation on publication of our work.

Reviewer #2 (Remarks to the Author):

The authors well answered the raised questions. They improved manuscript is suitable to be published in Nature communication. I recommend it to be published in this journal.

Reply: We thank the referee for his/her recommendation on publication of our work.

Reviewer #4 (Remarks to the Author):

Xu et al. reported negative interlayer magneto-resistance (NIMR) in BaGa₂ and interpreted their observation by the model of tunneling between Dirac fermions in the quantum limit. The technical part of the paper is well-written. The authors described their data and analyses clearly. I have also read the replies to the review report. I appreciate the authors' careful replies. Therefore, I would recommend the paper for publication. I have the following suggestions/comments.

Reply: We thank the referee for taking time to review the manuscript and recommend the paper for publication on Nature Communications. We really appreciate his/her insightful and valuable suggestions/comments which are very helpful in further improving our manuscript. Below, we present the point-to-point response to the referee.

1. It seems that the authors started to talk about the NIMR and “tunneling between Dirac fermions in the quantum limit” from the very beginning by citing previous works. No qualitative physical pictures were given. Given the broad audience of the Nature journals, would encourage the authors to make some schematics in a figure and also have a few sentences describing the physical picture.

Reply: We thank the referee for this valuable suggestion/comment. A schematic illustration of the interlayer transport mechanism and related description are added in the revised manuscript. Figure 1 is revised accordingly and shown below.

FIG. 1. Crystal structure and in-plane transport properties of BaGa₂. a Crystal structure and schematic of the interlayer tunneling of zeroth LLs's Dirac fermions of BaGa₂. The red and blue balls represent Ga and Ba atoms, respectively. b Single crystal X-ray diffraction pattern. Inset: the picture of BaGa₂ crystal. c Powder X-ray diffraction pattern with refined lattice parameters $a = 4.43\text{\AA}$ and $c = 5.08\text{\AA}$ (SG: P6/mmm). d Temperature dependence of in-plane $\rho_{xx}(T)$ and interlayer $\rho_{zz}(T)$. Inset: field dependence of Hall resistivity ρ_{xy} at 2 K. (e) In-plane MR at different temperatures.

2. While the technical descriptions are clear, I didn't get from reading the manuscript why observing the NIMR and "tunneling between Dirac fermions in the quantum limit" would be novel or significant. I would encourage the authors to do a better job in terms of convey the novelty or significance of their results.

Reply: We thank the referee for this insightful and valuable suggestion/comment. In this work, we report the negative interlayer magnetoresistance resulting from the tunneling of Dirac fermions between the zeroth Landau levels in BaGa₂, which is an interesting feature at the quantum limit. Our findings highlight the unusual role of the zeroth LLs in interlayer magnetotransport and enrich the novel transport properties of topological materials at the quantum limit. Some descriptions on the novelty and significance of this work is added in the revised manuscript.